**Data Availability Statement:** All relevant data are within the paper and its Supporting Information files.

# Factors associated with early pregnancy anemia in rural Sri Lanka: Does being 'under care' iron out socioeconomic disparities?

Gayani Shashikala Amarasinghe[1], Thilini Chanchala Agampodi[1], Vasana Mendis[2], Suneth Buddhika Agampodi[1,3]*

1 Department of Community Medicine, Faculty of Medicine and Allied Sciences, Rajarata University of Sri Lanka, Saliyapura, Sri Lanka, 2 Department of Pathology, Faculty of Medicine and Allied Sciences, Rajarata University of Sri Lanka, Saliyapura, Sri Lanka, 3 Department of Internal Medicine, Division of Infectious Diseases, Yale School of Medicine, New Haven, CT, United States of America

* suneth.agampodi@yale.edu

## Abstract

Globally, more than a third of pregnant women are anemic, and progress in its prevention and control is slow. Sri Lanka is a lower-middle-income country with a unique public health infrastructure that provides multiple interventions across the lifecycle for anemia prevention, despite which anemia in pregnancy remains a challenge. Studying the factors associated with maternal anemia in this context would provide unique information on challenges and opportunities encountered as low-and-middle-income countries attempt to control anemia by improving health care coverage. All first-trimester pregnant women registered for antenatal care in the Anuradhapura district between July 2019 to September 2019 were invited to participate in the baseline of a cohort study. Interviewer-administered and self-completed questionnaires were used. Anemia was defined using a full blood count. A hierarchical logistic regression model was built to identify factors associated with anemia. Out of 3127 participants, 451 (14.4%) were anemic. According to the regression model (Chi-square = 139.3, p<0.001, n = 2692), the odds of being anemic increased with the Period of gestation (PoG) (OR = 1.07, 95% CI = 1.01–1.13). While controlling for PoG, age and parity, history of anemia (OR = 3.22, 95%CI = 2.51–4.13), being underweight (OR = 1.64, 95%CI = 1.24–2.18), having the last pregnancy five or more years back (OR = 1.57,95%CI = 1.15–2.15) and having used intrauterine devices for one year or more (OR = 1.63, 95% CI = 1.16–2.30) increased the odds of anemia. Breast feeding during the last year (OR = 0.66, 95%CI = 0.49–0.90) and having used contraceptive injections for one year or more (OR = 0.61,95% CI = 0.45–0.83) reduced the risk of anemia. Proxy indicators of being in frequent contact with the national family health program have a protective effect over the socioeconomic disparities in preventing early pregnancy anemia. Maintaining the continuum of care through the lifecycle, especially through optimizing pre and inter-pregnancy care provision should be the way forward for anemia control.

**Funding:** TCA received This research was supported by the Accelerating Higher Education Expansion and Development (AHEAD) Operation of the Ministry of Higher Education, Sri Lanka funded by the World Bank (DOR STEM HEMS [6026-LK/ 8743-LK]) https://ahead.lk/. The funders had no role in study design, data collection and analysis, decision to publish, or preparation of the manuscript.

**Competing interests:** The authors have declared that no competing interests exist.

## Introduction

Anemia results from inadequate red cell or hemoglobin concentration to maintain the optimum oxygen supply for tissues [1]. Anemia in pregnant women could result in significant transgenerational morbidity, mortality, and productivity loss [2–10]. Pregnant women are especially vulnerable to anemia for several reasons, including physiological changes, socioeconomic or cultural restrictions to achieve optimal nutrition, and increased nutritional demand during pregnancy [11]. In fact, 37% of pregnant women worldwide are anemic [12]. A significant proportion of them, especially those identified as anemic during the early stages of pregnancy, would have already had anemia or marginal hemoglobin values before pregnancy. Therefore, prevention of anemia in pregnancy is essentially linked with anemia control in reproductive-age women.

Global progress in reducing anemia prevalence among pregnant women has been plodding [13]. Why a significant proportion of pregnant women would remain vulnerable to anemia despite multiple interventions at different stages of the lifecycle is a question that needs answers. Identifying factors associated with anemia is as crucial as identifying the underlying etiologies for anemia when answering this question.

Studies have shown that healthcare utilization during pregnancy, including the reception of antenatal care [13, 14], nutrient supplements [15–17], deworming [18, 19], and utilization of general healthcare services such as family planning [20–23] significantly reduced anemia in pregnant or non-pregnant reproductive-age females. Individual dietary patterns such as increased meat, fish and egg consumption, and community-level food fortification programs are also associated with anemia prevention [13, 14]. The odds of being anemic decreases with increased maternal age, increased gap between pregnancies, and lower parity [13, 14, 24]. Across geographical regions, underweight women have an increased risk of anemia compared to normal or overweight women [13, 20, 25]. Education level, wealth, employment status, hygiene, and sanitary facilities are also associated with anemia [25–31].

Sri Lanka is a lower-middle-income country providing well-structured family health services free of charge at the point of delivery with an island-wide coverage [32]. Public health provision has a strong orientation towards the continuum of care across the life cycle [33]. Despite the country's continuous efforts, nearly a third of pregnant women are anemic, and anemia prevalence among first-trimesters women is 18.4% [33]. Early studies in Sri Lanka have shown that anemia was more prevalent among pregnant women with a high parity [34]. However, perhaps due to the success of the family planning programme, this association was not reflected in later studies(8). Pregnant women at either extremity of reproductive age have reported a higher prevalence of anemia and iron deficiency [35]. Iron folate supplementation prior to the booking visit did not prevent anemia [35]. Iron folate supplementation during pregnancy, incorporated into the national maternal care package reduced the risk of anemia [32, 36]. Dietary habits such as milk, tea, and red meat consumption failed to show an association with anemia in pregnancy, but higher consumption of green leaves and eggs was associated with a lower risk for anemia [37]. An association between anemia and income level also has been shown [37]. Studies on non-pregnant reproductive age females have shown that anemia was not associated with economic status or dietary habits such as vegetarianism, consuming tea within an hour of a meal, and consuming meat, fish, and eggs [38]. Age, parity, and educational level were associated with anemia [39, 40]. Local studies reporting data on factors associated with anemia have employed heterogeneous study samples impairing the ability to generate an in-depth understanding of the determinants of maternal anemia, which would be invaluable to design further anemia control strategies. Therefore, we conducted this study to identify the factors associated with early pregnancy anemia in a large community-based

sample of pregnant women from a single geographical area in the country. Identifying associated factors enabled recognizing gaps in the current preventive strategies, alerting stakeholders to refine them to achieve better outcomes.

## Materials and methods

We carried out this descriptive cross-sectional study as a component of the baseline assessment of the Rajarata pregnancy cohort [41]. All the first trimester (Period of gestation less than 13 weeks) pregnant women registered in the pregnant mothers' registers of public health midwives in Anuradhapura district between July and September 2019 were invited to participate in the study. Anuradhapura is the geographically largest of the 25 administrative districts in the country. For the participant recruitment, special clinics were conducted weekly or fortnightly in each of the 22 medical officers of health (MOH) areas in the district. Anemia prevalence in the recruited sample of first trimester women (n = 3127) was 14.4% (n = 451), and iron deficiency, minor hemoglobinopathies, and vitamin B12 deficiency were identified as the major etiological contributors to anemia [42].

An interviewer-administered questionnaire was used to obtain demographic and health-related data. Menstrual blood loss was assessed using a pictogram. Health records, including the current and previous pregnancy records, and high-performance liquid chromatography (HPLC) for thalassemia screening were checked to verify the information. Socioeconomic data and data on dietary habits were obtained using self-completed questionnaires. All the tools were developed specifically for the study and were translated into the local languages (Sinhala and Tamil). They were validated by a panel of multidisciplinary experts including a consultant community physician, social epidemiologist, hematologist, physicians and public health midwives working in the area where study is conducted. Tools were pretested among 20 pregnant women with the same eligibility criteria but registered in the maternal care program prior to the study period. The questions were amended according to the results of pretesting in order to make sure they were comprehendible by participants. An interviewer guide was followed when administrating the questionnaire. Anthropometric measurements of participants were obtained following a standard protocol. A full blood count was performed on all the participants from a public health research laboratory with external and internal quality control methods [43]. A standardized protocol was followed when collecting, transporting and storing blood samples. A detailed account of the study is presented elsewhere [44]. All the questionnaires and protocols for anthropometric measurements and sample collection are available online [43].

### Measures

Anemia was defined as a hemoglobin level less than 11 g/dl. Minor hemoglobinopathies were identified based on the red cell index (microcytic anemia with a high red cell count), which showed a 100% positive predictive value in the HPLC tested subsample [42]. Independent variables included demographic factors, reproductive and gynecological factors, uptake of health services, diet, nutritional status, current health status and socio-economic conditions. During the completion of the interviewer-administered questionnaire, the expected date of delivery was verified with ultrasound confirmation whenever possible, and this was reassessed during the follow-up of the cohort. PoG was calculated using the date of the last menstrual period (LRMP) reported by the woman, and ultrasound confirmed date of delivery. Obtained values were triangulated to decide the most appropriate value. Maternal age was calculated using the date of birth. Education level of the participant and her husband, number of family members living with the participant (family size), parity, number of children, gap between the previous

and current pregnancy and duration of using different contraceptive methods were recorded as ordinal values and categorized later. Among women with regular menstrual cycles, who use sanitary pads, menstrual blood loss per cycle was calculated using a scoring system on the data obtained from the pictogram [45]. This pictogram has been used in previous studies conducted in the same setting [46]. A score of 1, 5 and 20 given when each used sanitary pad was lightly, moderately or completely stained when changed, respectively and an additional score of 5 was added if clots are passed [45]. Average blood loss per year was calculated based on the cycle length. This variable was binned into three equal percentiles categorizing the blood loss as low, medium and high. A score out of five was allocated to the frequency of consuming food items given in the questionnaire (ranging from1 for never consuming to 5 for consuming daily). Scores were added up as necessary (e.g.: scores for consuming chicken, other meat, freshwater fish, sea fish, canned fish, sprats, and dry fish were combined for the variable 'Frequency of consuming meat/fish'). The combined score was binned into three based on equal percentiles to determine low, moderate, and high consumption levels. BMI below 18.5 kg/m$^2$ was considered underweight and BMI of 25 kg/m$^2$ was considered overweight/ obese category [47]. Monthly income was also binned into four based on equal percentiles. Proxy indicators; using a water sealed toilet, having electricity at home, having a mobile phone, using biomass fuel for cooking, drinking water filtered using Reverse Osmosis, having a vehicle at home were used to represent socio-economic status.

### Data analysis

Data were analyzed using SPSS version 22. The data set used for the analysis is attached as S1 File. Chi square tests were performed to identify variables significantly associated with anemia. Based on the results of this bivariate analysis a two-step hierarchical logistic regression was performed to identify factors associated with anemia among first-trimester pregnant women. The PoG in weeks, maternal age (years), and parity (primy or multi) were added in block one. Having been breastfeeding within the past one year, ever having had anemia, BMI (those who were not underweight as reference category), having a vehicle at home, the gap between last and current pregnancies (reference category was being a primy or having the last pregnancy within five years), having used contraceptive injections for one year or more and having used intrauterine contraceptive devices for one year or more were added to the model as the block two. The final model was statistically significant (Chi-square = 139.3, p<0.001, n = 2692), which could explain 5.0% (Cox and Snell R square) and 9.2% (Nagaelkerke R squared) of the variance and correctly classified 86.2% of cases.

### Ethical clearance

This study was approved by the ethics review committee of the Faculty of Medicine and Allied Sciences, Rajarata University of Sri Lanka. Informed written consent was obtained from all participants. Proxy consent of guardians and ascent of participants were also obtained when the participant was below 18 years of age.

### Results

Out of the 3127 first trimester pregnant women recruited, 215 (7.5%) were teenage (below 20 years of age) and majority were ethnic Sinhalese (n = 2730, 87.1%), followed by Moor (n = 358, 11.4%) and Tamil (n = 39,1.2%), respectively. The majority (N = 1873, 60.3%) were educated up to the GCE ordinary level. Of the participants, 1071 (34.3%) were primi gravid, and 1158 (37.1%) had one child. Monthly family income varied between 2000 rupees (9.9 USD) to 0.5 million rupees (2469 USD). The median was 40,000 rupees (197.6 USD).

Underweight, normal, overweight and obesity were seen in 504 (16.6%), 993 (32.7%), 485 (16.0%) and 1053 (34.7%) respectively. The mean period of gestation (PoG) was eight weeks, and hemoglobin distribution according to the gestation revealed a reducing trend after ten weeks PoGs (Fig 1).

Five hundred sixty-one participants (18%) had a history of anemia, of whom 36.2% (n = 161) were anemic currently. Eighty-two out of the 561 had undergone HPLC testing, and 38 (5.9%) were confirmed as thalassemia trait. Only 32 out of the confirmed thalassemia trait participants were anemic during the first-trimester assessment. A quarter (n = 40) of anemic pregnant women with a history of anemia had a high Red Cell Index suggestive of minor hemoglobinopathy. Among the anemic pregnant women, one-third of ethnic Moore women (n = 19) had a high red cell index compared to only 18.2% (n = 72) of other women. Respectively 198, 360, 39, and 52 women with anemia had received dietary advice, oral iron, intravenous iron, and other vitamins as treatment for anemia at the time of the previous diagnosis.

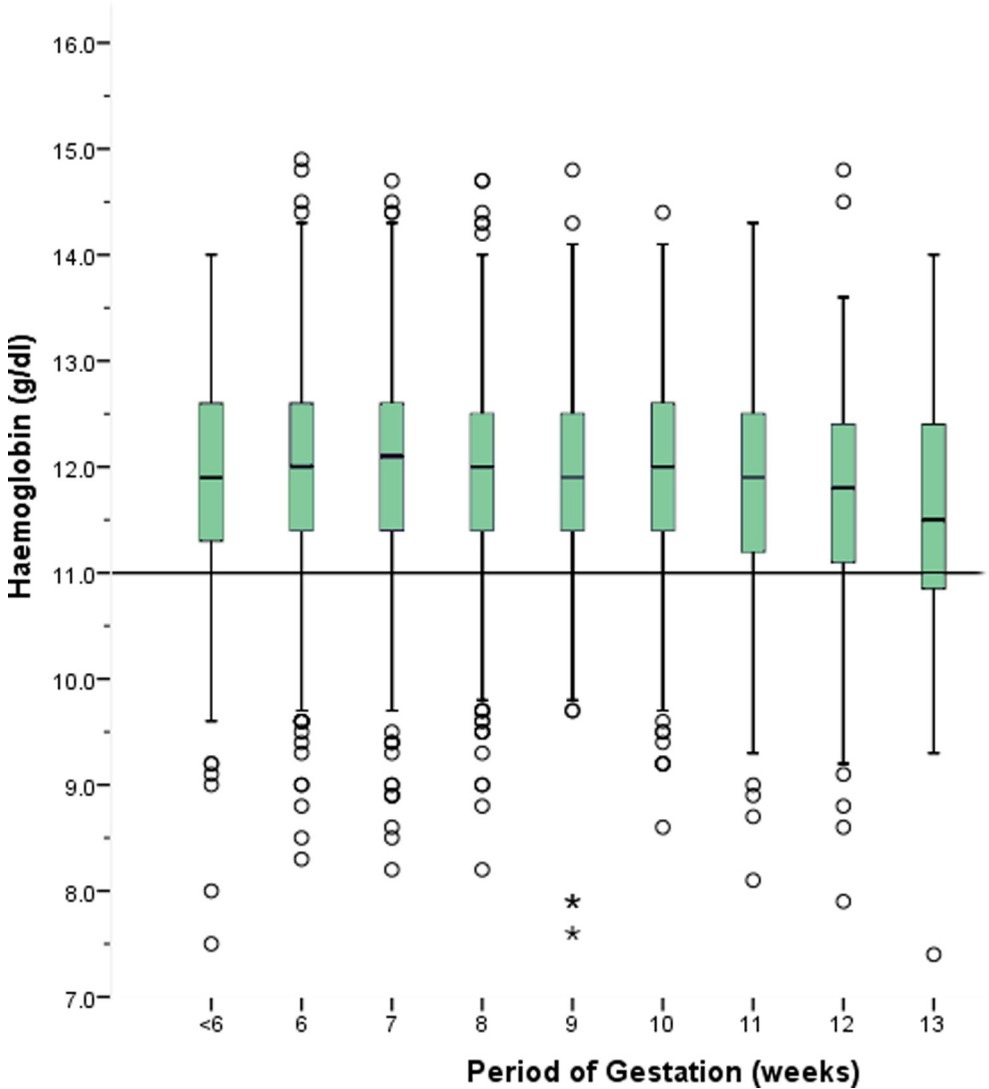

**Fig 1. Haemoglobin level of first-trimester pregnant women by the period of gestation.**

Seventeen had received blood transfusions, of which 11 were during the antenatal period of a previous pregnancy. Only 156 had evidence of normalized hemoglobin levels after initial treatments.

Chi-square tests showed that the anemia prevalence were statistically significantly different between groups categorized according to the gap between current and last pregnancy, body mass index (BMI) category, history of anemia, having used an intrauterine contraceptive device (IUCD) for one year or more, having used depo-medroxyprogesterone acetate injection (DMPA) for a year or more, breastfeeding during the past year and frequency of consuming dairy products (Table 1). Among studied socioeconomic factors, having a vehicle at home was significantly associated with anemia (Table 2).

Hierarchical logistic regression was performed to assess the impact of several factors on the likelihood of anemia in the first trimester of pregnancy (Table 3). After controlling for PoG at hemoglobin measurement, maternal age, and parity, the greatest contribution to the model was from having a history of anemia. Those with a history of anemia were 3.2 (P<0.001) times more likely to have anemia during the first trimester of pregnancy. Odds of having anemia increased with increasing PoG. Using IUCD for one year or more also increased the odds of having anemia by 1.6 times compared to those who had not used IUCD for a year or more. Having used DMPA injections for one year or more was protective against anemia. Those who were breastfeeding or had breastfed within the past year were less likely to be anemic than those who had never breastfed/had stopped breastfeeding more than a year ago. Underweight pregnant women were 1.6 times more likely to be anemic than pregnant women with normal BMI, overweight, or obese. Pregnant women who had had their last pregnancy five or more years back were 1.6 times more likely to have anemia compared to primiparous women and those who had their last pregnancy within five years.

## Discussion

This large community-based study provides a detailed analysis of probable associations of early pregnancy anemia in Sri Lanka, including demographic, nutritional, reproductive, and socioeconomic factors. In contrast to many studies in global literature, we have adopted a demarcation between these underlying associations and etiological causes for anemia, latter being described in detail elsewhere [42]. This analysis shows that PoG, history of anemia, being underweight, having the last pregnancy five or more years back, having used IUCD or DMPA, and breastfeeding during the last year were associated with anemia in early pregnancy. The study reflects that anemia during the first trimester of pregnancy represents the impact of pre-pregnancy determinants, including lifestyle, nutrition, and reproductive choices on the health of pregnant women. Strengthening the lifecycle approach in providing care for reproductive and other health issues would successfully address many factors associated with early pregnancy anemia in this community.

Review of global literature shows that factors associated with anemia in pregnancy are diverse and depend on the context [14]. Health and social inequities play a central role in the occurrence of anemia [13, 48]. Sri Lanka is unique in the global health landscape as it has a good public health infrastructure, well-established, accessible and affordable maternal care services with high coverage, and better gender equity compared to countries with similar economic characteristics [49, 50]. Studying the factors associated with maternal anemia in this context would provide unique information on opportunities and challenges that could be anticipated as low- and middle-income countries move forward with anemia reduction goals by improving health care coverage.

**Table 1. Demographic and health factors associated with anemia among first-trimester pregnant women in the Anuradhapura district.**

| Variable | | Anemic | | Non-anaemic | | | |
|---|---|---|---|---|---|---|---|
| | | N | % | N | % | X² | p |
| **Demographic factors** | | | | | | | |
| Age | <20 | 36 | 7.9 | 169 | 6.3 | | |
| | 20–29 | 248 | 54.2 | 1459 | 54.5 | | |
| | 30–39 | 158 | 34.6 | 1000 | 37.3 | 5.88 | 0.12 |
| | ≤40 | 15 | 3.3 | 51 | 1.9 | | |
| Education level | Up to O/L | 274 | 61.3 | 1587 | 59.9 | 0.31 | 0.58 |
| | Beyond O/L | 173 | 38.7 | 1062 | 40.1 | | |
| Partner's education level | Up to O/L | 307 | 68.2 | 1800 | 68.2 | 0.00 | 0.99 |
| | Beyond O/L | 143 | 31.8 | 840 | 31.8 | | |
| Family size | <5 | 294 | 65.6 | 1811 | 68.3 | 1.30 | 0.25 |
| | ≥5 | 154 | 34.4 | 839 | 31.7 | | |
| Ethnicity | Sinhala | 393 | 86.0 | 2331 | 87.2 | 0.97 | 0.60 |
| | Moore | 58 | 12.7 | 300 | 11.2 | | |
| | Other (Malay, Burger and other) | 6 | 1.3 | 42 | 1.6 | | |
| **Reproductive and gynaecological factors** | | | | | | | |
| Number of children | Non | 143 | 31.2 | 817 | 30.5 | 0.34 | 0.84 |
| | One | 147 | 32.5 | 849 | 31.8 | | |
| | Two or more | 164 | 36.3 | 1007 | 37.7 | | |
| Planned pregnancy (current) | Yes | 326 | 72.3 | 1950 | 73.9 | 0.55 | 0.46 |
| | No | 125 | 27.7 | 687 | 26.1 | | |
| The gap between current and last pregnancy | <5 years | 147 | 32.7 | 1084 | 40.8 | 13.29 | 0.001 |
| | ≥5 years | 160 | 35.6 | 754 | 28.4 | | |
| | Primiparous | 143 | 31.8 | 817 | 30.8 | | |
| Breast feeding currently/within last one year | No | 301 | 69.8 | 1590 | 62.4 | 8.85 | 0.003 |
| | Yes | 130 | 30.2 | 959 | 37.6 | | |
| Have had miscarriages ever | No | 230 | 74.4 | 1343 | 72.8 | 0.36 | 0.55 |
| | Yes | 79 | 25.6 | 502 | 27.2 | | |
| Has regular menstrual cycles | Yes | 388 | 85.5 | 2229 | 83.5 | 1.12 | 0.29 |
| | No | 66 | 14.5 | 441 | 16.5 | | |
| Average menstrual blood loss (before pregnancy) | Low | 138 | 32.5 | 865 | 33.7 | 0.96 | 0.62 |
| | Moderate | 138 | 32.5 | 864 | 33.7 | | |
| | High | 149 | 35.1 | 838 | 32.6 | | |
| **Health Service Uptake** | | | | | | | |
| Has used injectable contraceptives for ≥1year | No | 369 | 80.7 | 1997 | 74.5 | 8.17 | 0.004 |
| | Yes | 88 | 19.3 | 683 | 25.5 | | |
| Has used intrauterine contraceptive devices for ≥1year | No | 389 | 85.1 | 2422 | 90.4 | 11.57 | 0.001 |
| | Yes | 68 | 14.9 | 258 | 9.6 | | |
| Had taken anthelminthic drugs within last 6 months | Yes | 195 | 43.6 | 1151 | 43.5 | 0.00 | 0.96 |
| | No | 252 | 56.4 | 1495 | 56.5 | | |
| Attended pre pregnancy sessions | Yes | 71 | 22.5 | 431 | 23.1 | 0.05 | 0.82 |
| | No | 244 | 77.5 | 1434 | 76.9 | | |
| Has consumed iron supplements during last pregnancy | Yes | 269 | 91.5 | 1625 | 92.9 | 0.69 | 0.41 |
| | No | 25 | 8.5 | 125 | 7.1 | | |
| Has consumed iron supplements during postpartum period of last pregnancy | Yes, regularly | 178 | 61.6 | 1067 | 61.6 | 0.00 | 1.00 |
| | Yes, not regularly | 28 | 9.7 | 168 | 9.7 | | |
| | No | 83 | 28.7 | 496 | 28.7 | | |

*(Continued)*

**Table 1.** (Continued)

| Variable | | Anemic | | Non-anaemic | | | |
|---|---|---|---|---|---|---|---|
| | | N | % | N | % | $X^2$ | p |
| Consumed folic acid supplements before pregnancy | Yes | 266 | 81.8 | 1559 | 80.4 | 0.37 | 0.54 |
| | No | 59 | 18.2 | 380 | 19.6 | | |
| Consuming folic acid supplements currently | Yes | 421 | 93.3 | 2521 | 94.7 | 1.36 | 0.24 |
| | No | 30 | 6.7 | 141 | 5.3 | | |
| **Diet** | | | | | | | |
| Frequency of consuming meat/fish | Low | 64 | 35.2 | 466 | 42.4 | 3.38 | 0.18 |
| | Moderate | 67 | 36.8 | 366 | 33.3 | | |
| | High | 51 | 28.0 | 268 | 24.4 | | |
| Frequency of consuming milk/dairy products | Low | 85 | 46.7 | 378 | 34.5 | 10.27 | 0.006 |
| | Moderate | 59 | 32.4 | 452 | 41.2 | | |
| | High | 38 | 20.9 | 267 | 24.3 | | |
| **Nutritional status** | | | | | | | |
| History of anemia | No | 284 | 63.8 | 2210 | 83.9 | 104.52 | 0.001 |
| | Yes, during a past pregnancy/ postpartum | 109 | 24.5 | 319 | 12.1 | | |
| | Yes, not during a past pregnancy | 52 | 11.7 | 106 | 4.0 | | |
| Waist-to-hip ratio | ≥88 cm | 282 | 66.2 | 1707 | 67.4 | 0.25 | 0.62 |
| | <88 cm | 144 | 33.8 | 825 | 32.6 | | |
| BMI | Underweight | 101 | 23.0 | 403 | 15.5 | 20.22 | 0.001 |
| | Normal | 152 | 34.6 | 841 | 32.4 | | |
| | Overweight/obese | 186 | 42.4 | 1352 | 52.1 | | |
| **Current health status** | | | | | | | |
| Had blood in stools within last 6 months | Yes | 15 | 3.4 | 111 | 4.2 | 0.64 | 0.42 |
| | No | 428 | 96.6 | 2534 | 95.8 | | |
| Had melena within last 6 months | Yes | 9 | 2.0 | 45 | 1.7 | 0.22 | 0.64 |
| | No | 438 | 98.0 | 2600 | 98.3 | | |

Anemia increased with PoG (OR = 1.07, 95% CI 1.01 to 1.13). This observation can at least partly be explained through hemodilution, which starts as early as six weeks of PoG [51]. Looking at this pattern, we could argue that using 11g/dl as the hemoglobin cutoff until 13 weeks of PoG may classify some pregnant women with a lower hemoglobin level purely due to physiological changes in pregnancy, as anemic. This claim could be further supported by having a number of normal peripheral blood film reports in anemic women of this sample and common symptoms of anemia having a low sensitivity [42, 52]. Such mothers will undergo unnecessary interventions such as additional investigations, increased iron dose, and other nutritional supplements and dietary modifications that may lead to increased psychological and economic burden. Therefore, the harms and benefits of redefining the anemia cutoff for early pregnancy should be studied further through prospective studies.

Controlling for PoG, maternal age, and parity, the strongest predictor of first-trimester anemia, was having had anemia earlier in life. Non-modifiable etiologies such as thalassemia trait and membrane disorders contribute to a significant proportion of anemia in Sri Lankan communities [42, 53, 54]. On the other hand, treating anemia without a proper follow-up and etiological diagnosis may also lead to unresolved or recurrent anemia in women, which may be carried on to the pregnancy. More than two-thirds of the women with a history of anemia had been diagnosed during a previous pregnancy. Hence paying attention to ascertain the etiology

**Table 2. Social and economic conditions associated with anemia among first-trimester pregnant women in the Anuradhapura district.**

| | | Anemic | | Non-anaemic | | X² | p |
|---|---|---|---|---|---|---|---|
| | | N | % | N | % | | |
| **Alcohol and tobacco use** | | | | | | | |
| Have/had the habit of smoking tobacco products | Yes | 2 | 0.5 | 7 | 0.3 | 0.43 | 0.512 |
| | No | 398 | 99.5 | 2347 | 99.7 | | |
| Have/ Had the habit of betel (*Piper betle*) chewing (ever) | Yes | 13 | 3.2 | 43 | 1.8 | 3.46 | 0.06 |
| | No | 388 | 96.8 | 2313 | 98.2 | | |
| Has someone who smokes at home | Yes | 79 | 19.8 | 413 | 17.8 | 0.91 | 0.34 |
| | No | 321 | 80.3 | 1912 | 82.2 | | |
| Have/ever had the habit of consuming alcoholic beverages | Yes | 3 | 0.8 | 17 | 0.8 | 0.01 | 0.920 |
| | No | 362 | 99.2 | 2185 | 99.2 | | |
| **Self-reported abuse** | | | | | | | |
| Have you been emotionally or physically abused ever | Yes | 18 | 4.5 | 80 | 3.4 | 1.35 | 0.25 |
| | No | 379 | 95.5 | 2293 | 96.6 | | |
| **Proxy indicators of socio-economic status** | | | | | | | |
| Uses a water sealed toilet | Yes | 401 | 89.5 | 2438 | 92.1 | 3.30 | 0.07 |
| | No | 47 | 10.5 | 210 | 7.9 | | |
| Has electricity at home | Yes | 435 | 97.3 | 2605 | 98.4 | 2.69 | 0.10 |
| | No | 12 | 2.7 | 42 | 1.6 | | |
| Has a mobile phone | Yes | 438 | 97.8 | 2613 | 98.7 | 2.46 | 0.12 |
| | No | 10 | 2.2 | 34 | 1.3 | | |
| Uses biomass fuel for cooking | Yes | 201 | 44.9 | 1314 | 49.6 | 3.44 | 0.06 |
| | No | 247 | 55.1 | 1335 | 50.4 | | |
| Drinks water filtered using Reverse Osmosis | Yes | 312 | 69.8 | 1872 | 70.7 | 0.14 | 0.71 |
| | No | 135 | 30.2 | 777 | 29.3 | | |
| Has a vehicle at home | Yes | 391 | 89.1 | 2401 | 92.0 | 4.28 | 0.038 |
| | No | 48 | 10.9 | 208 | 8.0 | | |
| **Income** | | | | | | | |
| Monthly family income category | 1 (Lowest) | 50 | 28.4 | 280 | 26.4 | 1.55 | 0.67 |
| | 2 | 37 | 21.0 | 263 | 24.8 | | |
| | 3 | 41 | 23.3 | 255 | 24.1 | | |
| | 4(Highest) | 48 | 27.3 | 262 | 24.7 | | |

**Table 3. Hierarchical logistic regression explaining anemia in first-trimester pregnant women in Anuradhapura district.**

| | B | S.E. | Sig. | Odds ratio | 95% CI for OR | |
|---|---|---|---|---|---|---|
| | | | | | Lower | Upper |
| Period of gestation (weeks) | 0.07 | 0.03 | 0.018 | 1.07 | 1.01 | 1.13 |
| Maternal age (years) | -0.01 | 0.01 | 0.408 | 0.41 | 0.97 | 1.02 |
| Parity (primy or multi) | -0.06 | 0.20 | 0.782 | 0.95 | 0.64 | 1.41 |
| Having breast fed within one year | -0.41 | 0.16 | 0.009 | 0.66 | 0.49 | 0.90 |
| History of anaemia | 1.17 | 0.13 | 0.001 | 3.22 | 2.51 | 4.13 |
| Underweight (reference group- Normal/ overweight/ obese) | 0.5 | 0.15 | 0.001 | 1.64 | 1.24 | 2.18 |
| No vehicle at home | 0.32 | 0.19 | 0.099 | 1.37 | 0.95 | 2.00 |
| Used DMPA one year or more | -0.49 | 0.18 | 0.002 | 0.61 | 0.45 | 0.83 |
| Used IUCD one year or more | 0.49 | 0.18 | 0.005 | 1.63 | 1.16 | 2.30 |
| Last pregnancy 5 or more years back (reference group–primy/ last pregnancy within 5 years) | 0.45 | 0.16 | 0.005 | 1.57 | 1.15 | 2.15 |

of anemia identified in pregnant women and following them up during the postnatal period would be beneficial in view of providing a continuum of care.

Compared to primiparous women and women who had a pregnancy within the last five years, having had the last pregnancy five or more years back was associated with an increased risk for anemia. A study among non-pregnant women has also shown a similar trend of increased prevalence of anemia with the increased pregnancy gap [55]. Similarly, having breastfed a baby within the last year is also protective of anemia after controlling for age and pregnancy gap. These findings contrast with previous global literature showing that breast-feeding and lower pregnancy gaps increase the risk for anemia in women [13, 14]. A possible contextual explanation for this would be that recent contact with the public health system could be beneficial for maintaining the nutritional status of women in this community. Newly married women can attend the periconceptional clinics and receive nutritional supplementations and advice. Nutritional supplements are continued from the antenatal period to six months postpartum. Having a child under five years of age also brings the family to frequent close contact with the public health system. Attention given to women and children during the first few years of delivery through domiciliary and clinic care provided by the primary health care personnel is high in Sri Lanka. In the context where interpregnancy care is not strong or well structured, women who do not have frequent contact with the public health system may be at a disadvantage [33].

Contraceptive choices were associated with anemia, probably through alterations to menstrual blood loss. Hormonal contraceptives like DMPA have been shown to reduce anemia risk [23]. DMPA can cause temporary cessation of menstruation in many users [56]. Reduced menstrual loss may allow women to maintain their hemoglobin and iron levels. Copper IUCD provided by the national family planning program is the commonly used type of IUCD, which is known to increase menstrual blood loss and reduce iron stores in women [57, 58]. This may predispose women to develop anemia in the first trimester of pregnancy, especially when it occurs on top of already depleted nutritional stores or minor hemoglobinopathy variants. As IUCD is considered an ethnically appropriate modern method due to the absence of hormonal effects and is used by 10% of reproductive-age females in the country, it would be valuable to investigate the above observation further [59]. Providing adequate follow-up for contraceptive users and prompt treatment of excessive menstrual bleeding associated with contraceptive methods could help reduce anemia in reproductive-age females leading to a reduction of anemia in early pregnancy.

Comparable to global evidence [13], underweight pregnant women had 1.64 (95% CI 1.24–2.18) times more risk of being anemic than pregnant women with normal BMI. This signifies the association between anemia and nutrition during pre-pregnancy periods. A study among primary school children from the same setting has also shown an association between BMI and anemia, indicating that anemia and macro-nutrient deficiencies may be linked through common causative factors [60]. However, the income level or assets such as vehicles were not statistically significantly associated with anemia in the current study.

The study was conducted in a rural district of the country. The study's internal validity is high as the total study population was invited to participate, and the participation rate was high (86%). However, inter-district variations in the epidemiological patterns of anemia should be considered when generalizing the results to the country or beyond.

## Conclusion

Looking at the reproductive and nutrition-related associations to early pregnancy anemia in the current study results, we hypothesize that being in frequent contact with the national

family health program succeeds over socioeconomic disparities in preventing anemia in early pregnancy. Therefore, quantitative and qualitative improvement of already implemented nutritional interventions is important, and attention should be paid to maintaining them even during crises like the SARS Covid-19 pandemic. Even though the national program focuses on a continuum of care across the life cycle, interpregnancy care and care for eligible couples are still not strong. Strategies to mend this gap will reduce the burden of anemia in pregnancy through assessment and optimization of the macro and micronutrient status of women prior to pregnancy.

## Supporting information

**S1 File. Data used for the analysis.**
(XLS)

## Acknowledgments

We would like to acknowledge the RaPCo team and the public health staff in the Anuradhapura district.

## Author Contributions

**Conceptualization:** Gayani Shashikala Amarasinghe, Thilini Chanchala Agampodi, Vasana Mendis, Suneth Buddhika Agampodi.

**Data curation:** Suneth Buddhika Agampodi.

**Formal analysis:** Gayani Shashikala Amarasinghe.

**Funding acquisition:** Thilini Chanchala Agampodi.

**Investigation:** Gayani Shashikala Amarasinghe, Thilini Chanchala Agampodi, Vasana Mendis, Suneth Buddhika Agampodi.

**Methodology:** Gayani Shashikala Amarasinghe, Thilini Chanchala Agampodi, Vasana Mendis, Suneth Buddhika Agampodi.

**Project administration:** Gayani Shashikala Amarasinghe, Thilini Chanchala Agampodi, Suneth Buddhika Agampodi.

**Supervision:** Thilini Chanchala Agampodi, Vasana Mendis, Suneth Buddhika Agampodi.

**Writing – original draft:** Gayani Shashikala Amarasinghe.

**Writing – review & editing:** Thilini Chanchala Agampodi, Vasana Mendis, Suneth Buddhika Agampodi.

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
