## [Decision Letter · Decision Letter 0]

8 Jun 2022

PONE-D-22-01123Factors associated with early pregnancy anemia in rural Sri Lanka: Does being 'under care' iron out socioeconomic disparities?PLOS ONE

Dear Dr. Agampodi,

Thank you for submitting your manuscript to PLOS ONE. After careful consideration, we feel that it has merit but does not fully meet PLOS ONE’s publication criteria as it currently stands. Therefore, we invite you to submit a revised version of the manuscript that addresses the points raised during the review process.

Please submit your revised manuscript by Jul 23 2022 11:59PM. If you will need more time than this to complete your revisions, please reply to this message or contact the journal office at plosone@plos.org. Please include the following items when submitting your revised manuscript:A rebuttal letter that responds to each point raised by the academic editor and reviewer(s). You should upload this letter as a separate file labeled 'Response to Reviewers'.A marked-up copy of your manuscript that highlights changes made to the original version. You should upload this as a separate file labeled 'Revised Manuscript with Track Changes'.An unmarked version of your revised paper without tracked changes. You should upload this as a separate file labeled 'Manuscript'.

We look forward to receiving your revised manuscript.

Kind regards,

Rubeena Zakar, Ph.D

Section Editor

PLOS ONE

Journal Requirements:

Reviewers' comments:

Reviewer's Responses to Questions

**Comments to the Author**

1. Is the manuscript technically sound, and do the data support the conclusions?

Reviewer #1: Yes

Reviewer #2: Yes

2. Has the statistical analysis been performed appropriately and rigorously? 

Reviewer #1: Yes

Reviewer #2: Yes

3. Have the authors made all data underlying the findings in their manuscript fully available?

Reviewer #1: Yes

Reviewer #2: Yes

4. Is the manuscript presented in an intelligible fashion and written in standard English?

Reviewer #1: Yes

Reviewer #2: Yes

5. Review Comments to the Author

Reviewer #1: Although the manuscript has merit, unfortunately there is abundance of literature published since 2018 on this topic and in similar settings. There is very little that this study adds to existing knowledge base. Examples of similar recent published studies include:

Amarasinghe GS, Agampodi TC, Mendis V, Malawanage K, Kappagoda C, Agampodi SB. Prevalence and aetiologies of anaemia among first trimester pregnant women in Sri Lanka; the need for revisiting the current control strategies. BMC pregnancy and childbirth. 2022 Dec;22(1):1-2.

Andersen CT, Cain JS, Chaudhery DN, Ghimire M, Higashi H, Tandon A. Assessing public financing for nutrition in Bhutan, Nepal and Sri Lanka. Maternal & Child Nutrition. 2022 Mar 21:e13320.

Abeywickrama HM, Koyama Y, Uchiyama M, Shimizu U, Iwasa Y, Yamada E, Ohashi K, Mitobe Y. Micronutrient status in Sri Lanka: a review. Nutrients. 2018 Nov;10(11):1583.

Reviewer #2: Dear author/s,

I have tried to assess your paper and it is a kind of problem-solving research and sounds good. But there are still some unclear issues in the paper and you are expected to work on them. The journal guidelines should be followed and you can find other comments and suggestions in the main text of your paper.

Regards,

Fufa

6. PLOS authors have the option to publish the peer review history of their article (what does this mean?). If published, this will include your full peer review and any attached files.

Reviewer #1: No

Reviewer #2: **Yes: **FUFA ABUNNA

---

## [Author Response · Author response to Decision Letter 0]

27 Jun 2022

Reviewer #2: Dear author/s,

I have tried to assess your paper and it is a kind of problem-solving research and sounds good. But there are still some unclear issues in the paper and you are expected to work on them. The journal guidelines should be followed and you can find other comments and suggestions in the main text of your paper.

Thank you very much for the comments. I have addressed all the concerns raised in the manuscript and a point-by-point reference to them are added below.

How is it possible to deal with a self-administered questionnaire? Are they professionals? I think it should have been made via interview.

Thank you for raising this concern. This community has a high (95%) level of literacy level[1]. In addition to that, participants who found it difficult to read and write were helped by data collectors. The self-administered questionnaire was simple and short with minimum writing and involvement. It was used to obtain data on mainly diet. Before administering we pilot tested the questionnaire and necessary amendments were made while it was made sure that it could be read and understood by participants. We included this in the manuscript LN-112- 113

“All the tools were developed specifically for the study and were translated into the local languages (Sinhala and Tamil). They were validated by a panel of multidisciplinary experts including a consultant community physician, social epidemiologist, hematologist, and physicians and public health midwives working in the area where the study is conducted. Tools were pretested among 20 pregnant women with the same eligibility criteria but registered in the maternal care program prior to the study period. The questions were amended according to the results of pretesting in order to make sure they were comprehendible by participants.”

(Having a vehicle at home)- Is it important for this research? 

Thank you for this point. Yes, having a vehicle at home and other indicators resembling household assets and housing conditions were used as proxy indicators for socio-economic status. We have experienced that asking about household income is not that reliable in studies conducted in this location possibly as many residents are full-time or part-time farmers with variable income over the months. So, assets are more reliable as a measure of economic status over time. It was included in the multivariate analysis as this variable showed a statistically significant association with anemia in the bivariable analysis. The table 1 was r splitted into two and reclassified to make this point clear. The point was mentioned in the newly added paragraph on measures LN-153-156

“Proxy indicators; using a water sealed toilet, having electricity at home, having a mobile phone, using biomass fuel for cooking, drinking water filtered using Reverse Osmosis, having a vehicle at home were used to represent socio-economic status.”

Which model?

Thank you. We meant the Regression model. This was clarified in the text. LN- 32

Keywords

Thank you for pointing this out. Keywords were inserted

Did u translate the questionnaire to the local language? Otherwise very difficult to understand some technical terminologies

Thank you very much. Yes. All the tools were translated into the two local languages, Sinhala and Tamil, and pretested. A clarification was added in the text. LN-112

“ All the tools were developed specifically for the study and were translated into the local languages (Sinhala and Tamil).”

A reference to tools available online was also added to the manuscript.

Full blood count - Is it done manually or using a machine?

It was done with a machine (Beckman 03 part hematological analyzer ) 

Teenage - define

Thank you for pointing this out. We added a clarification to the text. We considered below 20 years as teenage. LN - 179

Not clear – Fig 1 Hemoglobin distribution of first-trimester pregnant women by the period of gestation 

Thank you. The figure caption was changed as follows.

“Fig 1 Haemoglobin level of first trimester pregnant women by the period of gestation”

Which statistical software did u use?

We used SPSS version 22. This has been mentioned in the Methods section of the manuscript. LN-158

It is very difficult to understand this table, either divide the table into separate descriptions or reduce its contents. Above all, it lacks clarity, cut-off points, and definitions. It is better to use graphs, pie charts, and other methods of data presentations

Thank you very much for the suggestion. I have added a paragraph under the Measures to clarify the variables and definitions that have been used in this table. LN 128 - 156

“ Measures

Anemia was defined as a hemoglobin level less than 11 g/dl. Minor hemoglobinopathies were identified based on the red cell index (microcytic anemia with a high red cell count), which showed a 100% positive predictive value in the HPLC tested subsample(15). Independent variables included demographic factors, reproductive and gynecological factors, uptake of health services, diet, nutritional status, current health status and socio-economic conditions. During the completion of the interviewer-administered questionnaire, the expected date of delivery was verified with ultrasound confirmation whenever possible, and this was reassessed during the follow-up of the cohort. PoG was calculated using the date of last menstrual period (LRMP) reported by the woman, and ultrasound confirmed date of delivery and data were triangulated decide the most appropriate value. Maternal age was calculated using the date of birth. Education level of the participant and her husband, number of family members living with the participant (family size), parity, number of children, gap between the previous and current pregnancy and duration of using different contraceptive methods were recorded as ordinal values and categorized later. Among women with regular menstrual cycles, who use sanitary pads, menstrual blood loss per cycle was calculated using a scoring system on the data obtained from the pictogram (score of 1, 5 and 20 given when each used sanitary pad was lightly, moderately or completely stained when changed, respectively and 5 was added if clots are passed) (18). Average blood loss per year was calculated based on the cycle length. This variable was binned into three equal percentiles. A score out of five was allocated to the frequency of consuming food items given in the questionnaire (1 for never, 5 for daily). Scores were added up as necessary (e.g.: scores for consuming chicken, other meat, freshwater fish, sea fish, canned fish, sprats, and dry fish were combined for the variable ‘Frequency of consuming meat/fish’). The combined score was binned into three based on equal percentiles to determine low, moderate, and high consumption levels). BMI below 18.5 kg/m2 was considered underweight and BMI of 25 kg/m2 was considered overweight/ obese category(19). Monthly income was also binned into four based on equal percentiles. Proxy indicators; using a water sealed toilet, having electricity at home, having a mobile phone, using biomass fuel for cooking, drinking water filtered using Reverse Osmosis, having a vehicle at home were used to represent socio-economic status.”

The table was split into two for clarity. The first one includes demographic and health related variables whereas the second one includes social and economic variables.

How is age defined? From their ID of birth certificate? Better to reduce the categories 

Thank you for raising this issue. We collected the birth date of the participants and calculated the age for the date of data collection. This clarification was added to the methods section (see above). We also reduced the number of age categories in Table 1 and updated the statistics as suggested.

Why you mix these two ethnicities? Either make correction in the methods part otherwise it looks as if it is ethnic ‘cleansing’

Thank you pointing this out. We mixed it as the percentage having a minor hemoglobinopathy was higher among the Moore/ Malay ethnicity. Since it does not seem to be appropriate, I changed the categories as Sinhala, Moore, and other (Burgher, Malay, other). (Table 1) 

Has regular menstrual cycles- Is it not difficult to explain this.? How is irregularity defined? Do women have the experience of registering their menstrual cycle? I think this is more subjective. Either you have to set a cutoff point to define irregularity!

We assumed that women would have an understanding of their menstrual cycle is generally regular or irregular. These are women still in their first-trimester. So, they will remember their menstrual periods. This was not found problematic during the pretesting or data collection. Further it’s a routine question asked from pregnant women in getting the history in Sri lanka by both the Public Health Midwives and Medical Officers. Hence the question is well understood by the community and well received.

Average menstrual blood loss before pregnancy - Again this is more of subjective! What is it? The number of days or volume of blood?

Thank you for raising this issue. We used a pictogram to identify the extent of filling and the number of pads in the amount of filling. Then a calculation was made about the blood loss based on a previous publication[2]. This was multiplied by the number of periods over a year based on the cycle length. The calculation was made only for women with regular cycles and using sanitary pads. This was clarified in the measures section (please see above). This method is used in previous studies in the same population[3].

Dietary intake - Still these are more subjective. What is low, moderate or high or low? Please set a kind of cut-off Points.

Thank you for pointing out this deficiency. A detailed explanation of the variable was added to the paragraph on measures (please see above).

Folic acid supplements - How do they know these supplements?

Thank you for raising this concern. Folic acid supplements are provided free of charge through the public health midwife. Since health education is provided on this at preconception care and at the registration of pregnancy (participants are those registered for the maternal care program) and the supplement is given to mothers with the registration, they are more aware of it even by name. We also used the term “small white pill” to recognize the pill during the interview as it is the appearance of the tablet distributed by the medical supplies department at that time. 

Have/ Had the habit of betel chewing (ever) - what is it

Thank you for pointing this out. Betel leaves (piper betle) are chewed often with areca nuts, Lyme, tobacco and sometimes different herbs. I added the scientific nomenclature for clarity (Table 2).

Monthly family income – cutoff

Thank you for pointing this out. The description was added in the paragraph on measures.

It is not well discussed. It is better to focus on very important findings. What is new in this study? What makes this study special and different from other studies conducted elsewhere in the globe. 

Thank you for pointing this out. I have revised the discussion section to focus more on important findings. I added the following component to highlight the usefulness of the study. LN- 237 - 252

“Review of global literature shows that factors associated with anemia in pregnancy are diverse and depend on the context(20). Health and social inequities play a central role in occurrence of anemia(21,22). Sri Lanka is unique in the global health landscape as it has a good public health infrastructure, well established, accessible and affordable maternal care services with high coverage and better gender equity compared to countries with similar economic characteristics(23,24). Studying the factors associated with maternal anemia in this context would provide unique information on opportunities and challenges that could be anticipated as low- and middle-income countries move forward with anemia reduction goals by improving health care coverage.”

Reviewer #1: Although the manuscript has merit, unfortunately there is abundance of literature published since 2018 on this topic and in similar settings. There is very little that this study adds to existing knowledge base. Examples of similar recent published studies include:

Amarasinghe GS, Agampodi TC, Mendis V, Malawanage K, Kappagoda C, Agampodi SB. Prevalence and aetiologies of anaemia among first trimester pregnant women in Sri Lanka; the need for revisiting the current control strategies. BMC pregnancy and childbirth. 2022 Dec;22(1):1-2.

Andersen CT, Cain JS, Chaudhery DN, Ghimire M, Higashi H, Tandon A. Assessing public financing for nutrition in Bhutan, Nepal and Sri Lanka. Maternal & Child Nutrition. 2022 Mar 21:e13320.

Abeywickrama HM, Koyama Y, Uchiyama M, Shimizu U, Iwasa Y, Yamada E, Ohashi K, Mitobe Y. Micronutrient status in Sri Lanka: a review. Nutrients. 2018 Nov;10(11):1583.

Thank you for the comment. However, it seems that the reviewer has mistakenly assumed the content in this manuscript describe etiological factors (such as micronutrient deficiencies) which are commonly described in global literature under associated factors’. In fact, one of the papers given as example of similar literature describe the etiologies related to anemia in the same sample of women, published separately due to the clear distinction of two entities we adopted in conducting the research (as well as due to the scope being too large to be published as a single paper). This study describes an explanatory model for anemia in early pregnancy.

Studies looking at associated factors have clearly shown how diverse and context-specific they could be. We believe that a similar study has not been conducted in a lower-middle income setting where anemia prevalence remains relatively high despite a multitude of targeted interventions (please see reply to previous reviewer comment). Such a study will reflect the gaps of current strategies and points towards potential remedial actions, which is the uniqueness of the study. 

“Local studies reporting data on factors associated with anemia have employed heterogeneous study samples impairing the ability to generate an in-depth understanding of the determinants of maternal anemia, which would be invaluable to design further anemia control strategies. Therefore, we conducted this study to identify the factors associated with early pregnancy anemia in a large community-based sample of pregnant women from a single geographical area in the country.” LN 88-95

Review of global literature shows that factors associated with anemia in pregnancy are diverse and depend on the context(30). Health and social inequities play a central role in the occurrence of anemia(31,32). Sri Lanka is unique in the global health landscape as it has a good public health infrastructure, well-established, accessible, and affordable maternal care services with high coverage and better gender equity compared to countries with similar economic characteristics(33,34). Findings of this study provides unique information on opportunities and challenges that could be anticipated as low- and middle-income countries move forward with anemia reduction goals by improving health care coverage.” LN 249

Reference

1. Department of Census and Statistics Ministry of National Policies and Economic Affairs. Sri Lanka Demographic and Health Survey 2016. Colombo; 2017.

2. HIGHAM JM, O’BRIEN PMS, SHAW RW. Assessment of menstrual blood loss using a pictorial chart. Br J Obstet Gynaecol. 1990;97:734–9.

3. Chathurani U, Dharshika I, Galgamuwa D, Wickramasinghe N, Agampodi T, Nugegoda D, et al. Association between menstrual blood loss before pregnancy and microcytic hypochromic anemia during pregnancy among currently pregnant women in Anuradhapura district. 2014.

---

## [Decision Letter · Decision Letter 1]

1 Sep 2022

Factors associated with early pregnancy anemia in rural Sri Lanka: Does being 'under care' iron out socioeconomic disparities?

PONE-D-22-01123R1

Dear Dr. Agampodi,

We’re pleased to inform you that your manuscript has been judged scientifically suitable for publication and will be formally accepted for publication once it meets all outstanding technical requirements.

Kind regards,

Rubeena Zakar, Ph.D

Section Editor

PLOS ONE

Additional Editor Comments (optional):

Reviewers' comments:

Reviewer's Responses to Questions

**Comments to the Author**

1. If the authors have adequately addressed your comments raised in a previous round of review and you feel that this manuscript is now acceptable for publication, you may indicate that here to bypass the “Comments to the Author” section, enter your conflict of interest statement in the “Confidential to Editor” section, and submit your "Accept" recommendation.

Reviewer #1: All comments have been addressed

Reviewer #2: All comments have been addressed

2. Is the manuscript technically sound, and do the data support the conclusions?

Reviewer #1: Yes

Reviewer #2: Yes

3. Has the statistical analysis been performed appropriately and rigorously? 

Reviewer #1: Yes

Reviewer #2: Yes

4. Have the authors made all data underlying the findings in their manuscript fully available?

Reviewer #1: Yes

Reviewer #2: Yes

5. Is the manuscript presented in an intelligible fashion and written in standard English?

Reviewer #1: Yes

Reviewer #2: Yes

6. Review Comments to the Author

Reviewer #1: Thank you for addressing the comments highlighted in the previous review. The manuscript is now suitable for publication in the journal. This article will add to the existing knowledge base in this area within a LMIC settings.

Reviewer #2: (No Response)

7. PLOS authors have the option to publish the peer review history of their article (what does this mean?). If published, this will include your full peer review and any attached files.

Reviewer #1: No

Reviewer #2: **Yes: **FUFA ABUNNA

---

## [Editor Report · Acceptance letter]

26 Sep 2022

PONE-D-22-01123R1 

Factors associated with early pregnancy anemia in rural Sri Lanka: Does being ‘under care’ iron out socioeconomic disparities? 

Dear Dr. Agampodi:

I'm pleased to inform you that your manuscript has been deemed suitable for publication in PLOS ONE. Congratulations! Your manuscript is now with our production department. 

Kind regards, 

on behalf of

Dr. Rubeena Zakar 

Section Editor

PLOS ONE